# Quality of Life, Sarcopenia and Nutritional Status in Patients with Esophagogastric Tumors before and after Neoadjuvant Therapy

**DOI:** 10.3390/cancers16061232

**Published:** 2024-03-21

**Authors:** Lena Schooren, Grace H. Oberhoff, Alexander Koch, Andreas Kroh, Tom F. Ulmer, Florian Vondran, Jan Bednarsch, Ulf P. Neumann, Sophia M. Schmitz, Patrick H. Alizai

**Affiliations:** 1Department of Surgery, University Hospital Aachen, Pauwelsstr. 30, 52074 Aachen, Germany; lschooren@ukaachen.de (L.S.); grace.oberhoff@rwth-aachen.de (G.H.O.); akroh@ukaachen.de (A.K.); tomflorian.ulmer@uk-essen.de (T.F.U.); fvondran@ukaachen.de (F.V.); ulf.neumann@uk-essen.de (U.P.N.); 2Department of Internal Medicine, University Hospital Aachen, Pauwelsstr. 30, 52074 Aachen, Germany; akoch@ukaachen.de; 3Department of Surgery, University Hospital Essen, Hufelandstr. 55, 45147 Essen, Germany; jan.bednarsch@uk-essen.de; 4Department of Surgery, Gemeinschaftskrankenhaus Bonn, Prinz-Albert-Straße 40, 53113 Bonn, Germany; p.alizai@gk-bonn.de

**Keywords:** gastrointestinal cancer, sarcopenia, nutrition, HRQoL, quality of life

## Abstract

**Simple Summary:**

Aside from a decreased quality of life, patients with esophagogastric tumors often suffer from difficulties with eating that impact nutritional status. Both the presence of cancer and the impacted nutritional status can also reduce muscle mass. All of this can impact treatment decisions: a patient with low muscle mass and low caloric impact might have a lower tolerance for chemotherapy or other treatment courses. However, treating physicians do not always have immediate access to current data on muscle mass and nutritional status. If there was a correlation between quality of life and these parameters, this might allow for the estimation of real-time muscle mass and suggest further controlled studies.

**Abstract:**

(1) Background: Health-related quality of life (HRQoL) gains importance as novel treatment options for individuals with esophagogastric tumors to improve long-term survival. Impaired HRQoL has been shown to be a predictor of overall survival. Sarcopenia is a known prognostic factor for postoperative complications. As the regular control of sarcopenia through CT scans might not always be possible and HRQoL and nutritional scores are easier to obtain, this study aimed to assess the relationship between nutritional scores, HRQoL and skeletal muscle mass in patients undergoing chemotherapy for cancers of the upper gastrointestinal tract. (2) Methods: Eighty patients presenting with tumors of the upper GI tract were included and asked to fill out the standardized HRQoL questionnaire, EORTC’s QLQ-C30. Nutritional status was assessed using the MNA, MUST and NRS 2002 scores. Sarcopenia was determined semi-automatically based on the skeletal muscle index at the L3 vertebrae level in staging CT scans. (3) Results: In chemo-naïve patients, HRQoL summary scores correlated significantly with nutritional scores and SMI. SMI and HRQoL prior to neoadjuvant therapy correlated significantly with SMI after treatment. (4) Conclusions: HRQoL is a helpful tool for assessing patients’ overall constitution. The correlation of HRQoL summary scores and SMI might allow for a rough assessment of skeletal muscle status through HRQoL assessment in chemo-naïve patients.

## 1. Introduction

With a high prevalence mostly in Asian and African countries but also a notable presence in Europe, cancers of the stomach and the esophagus account for almost 10% of cancer cases worldwide [1,2]. With rapidly improving 5-year survival rates, the focus has turned to other parameters, like muscle mass, nutritional status or health-related quality of life (HRQoL) as an outcome parameter that is easily obtainable in almost any setting [3,4]. Sarcopenia is usually defined as a loss of skeletal muscle mass and muscle function and is generally regarded as a geriatric condition, but it can also be found in underlying illnesses like cancer cachexia [5,6]. This not only affects muscle function but can also concern overall quality of life. HRQoL is described by the European Organisation for Research and Treatment of Cancer as covering “the subjective perceptions of the positive and negative aspects of cancer patients’ symptoms, including physical, emotional, social, and cognitive functions and [..] disease symptoms and side effects of treatment” [7]. 

In other cancers, nutrition, muscle mass and HRQoL showed correlations and could even predict one another. In studies of head and neck, ovarian and bladder cancer, HRQoL was found to be prognostic for overall survival [8,9,10,11], and in patients receiving VATS lobectomy for lung cancer, a poor preoperative QoL was associated with a longer hospital stay after the procedure [12]. Meanwhile, malnutrition was associated with the cessation of treatment and lower overall survival in patients with non-small-cell lung cancer, as well as survival rates and systemic recurrence in biliary tract cancers and adverse outcomes after surgery for rectal cancer [13,14,15,16,17]. Sarcopenia was associated with a higher risk of developing treatment-associated toxicity in patients with head and neck cancer and a higher rate of postoperative complications in patients with colorectal cancer [18,19]. It was also predictive of lower survival rates for patients with lung cancer, colorectal cancer and bladder cancer [20,21,22]. Furthermore, sarcopenia was associated with lower QoL and depression symptoms in patients with advanced cancers, highlighting the importance of assessing such parameters regularly [23,24].

Tumors of the upper gastrointestinal (GI) tract are known to cause a reduction in HRQoL for patients both during and after oncological therapy due to symptoms caused by the tumor itself and, later, the rather aggressive treatment [25,26,27]. Aside from receiving insight into patients’ HRQoL to provide better all-around care, HRQoL has additionally been shown to be a predictor for overall survival (OS) in upper GI cancers [28,29]. One important factor that is assessed in standardized HRQoL questionnaires is physical function as perceived by the patients themselves. Patients are asked to indicate whether they experience trouble doing strenuous activities or taking long walks or whether they need to stay in bed or a chair during the day or need help with ordinary activities like eating or dressing. This might give insight into how much a patient is physically active independent from measured parameters like muscle mass or muscle strength. Physical activity has also been shown to have a positive correlation with HRQoL in survivors of cancer, additionally contributing to the importance of assessing and following up this specific scale [30].

Another aspect recently gaining importance as a clinical prognostic factor in upper GI cancers is sarcopenia, which has not only been shown to be prognostic for progress-free and overall survival [31,32,33,34,35,36,37,38], but also reportedly correlates with postoperative complications like pneumonia, and might even influence anastomotic leakage [31,35,39,40,41,42]. Specifically, loss of skeletal muscle during neoadjuvant treatment has been suggested to raise anastomotic leakage rates, while chemotherapy, according to the FLOT protocol, has been shown to raise sarcopenia rates from 16% to 33% [40,43]. 

Recent attention has been attributed to muscle function and strength as assessed by handgrip strength or physical exercise rather than the measurement of plain muscle mass [19,44,45,46]. On the other hand, nutritional scores, which are easy and fast to assess, have also been shown to correlate with OS, duration of chemotherapy and HRQoL [47,48,49,50,51,52]. Not only the QLQ-C30 summary score but also singular eating-related QLQ-OG25 symptom scores have been shown to be associated with OS [29]. Even though this clearly demonstrates the importance of the nutritional status prior to surgery, a German study by Pirlich et al. showed high malnutrition rates of 44% in patients undergoing abdominal surgery [53]. Both sarcopenia and low nutritional scores were recently shown to be associated with unplanned readmission after gastrectomy [54]. 

However, the regular control of muscle mass proves to be difficult. Factors like nutritional scores and HRQoL would be much easier to assess than CT-based sarcopenia, but there has been no extensive research on the association between those scores and sarcopenia in esophagogastric cancers. Therefore, the aim of this study was to assess the relationship between nutritional scores, HRQoL and perceived physical function and skeletal muscle mass in patients undergoing chemotherapy for cancers of the upper gastrointestinal tract before and after neoadjuvant treatment. 

## 2. Materials and Methods

### 2.1. Trial Design and Study Population

Between August 2020 and November 2022, patients who presented at RWTH Aachen university hospital’s surgical outpatient clinic with tumors of the upper GI tract (esophagus and stomach) were included in this cross-sectional study. Exclusion criteria were insufficient knowledge of the German language and an age below 18 years.

Patients provided written, informed consent for participation in the study. This study was approved by the ethics committee of RWTH Aachen University, Aachen, Germany (#EK 419/20).

Data of some patients in this patient group have been published previously as part of the research group’s work on upper GI cancers [55].

### 2.2. Data Collection

Patients that presented with tumors of the upper GI tract were included in this study and assigned to one of two groups depending on their treatment status. Patients that had not yet received chemo- or radiation therapy were assigned to the chemo-naïve group (from hereon t1), irrespective of their planned course of treatment. Patients that presented after neoadjuvant treatment were assigned to the post- chemo- or radiation therapy group (from hereon t2). Interviewing for this study included data collection for HRQoL and nutritional assessments as well as general patient information. HRQoL questionnaires were self-administered in order to minimize external influence [56]. For the determination of HRQoL, patients that presented in the surgical outpatient clinic were asked to self-report their HRQoL using printed German versions of the EORTC’s QLQ-C3 [57,58] questionnaire. The QLQ-C30 questionnaire consists of 30 questions about cancer-related functions and symptoms. For each question but two that assess overall health status and quality of life, patients were asked to answer on a scale of 1 to 4 based on agreement with the statement or question, ranging from “not at all” to “very much“. For the two remaining questions assessing overall health status and HRQoL, patients were given options from one to seven, with one indicating extremely low and seven meaning extremely high HRQoL and health status. Nutritional status was assessed using the MNA [59], MUST [60] and NRS 2002 [60] scores. 

Additionally, the skeletal muscle index at level L3 was retrieved from staging CT scans for both patient groups. If available, the SMI was determined before and after neoadjuvant therapy, regardless of patient group, to allow for the longitudinal analysis of SMI correlations. The skeletal muscle index (SMI) was determined based on staging CT scans using the 3D Slicer software version 4.11.20210226. First, skeletal muscle area (SMA) at the level of L3 was calculated, and then, it was divided by squared height to compute the SMI in the unit of cm^2^/m^2^ (example can be seen in Figure 1).

### 2.3. Statistical Analysis

Answers from the HRQoL questionnaire were processed according to the EORTC’s manual by combining them into six functional and nine symptom scales for the QLQ-C30 questionnaire and subsequently adapting them to a scale of 0 to 100 by applying linear transformation. 

Recent evidence suggests that the summary score, global QoL scale, and physical functioning scale are stronger predictors of all-cause mortality than the other functioning and symptom scales of the QLQ-C30 [28]. Thus, only these scores were used for the correlation analysis. The QLQ-C30 summary score was calculated as an added indicator of HRQoL. The calculation was carried out according to the official EORTC guidelines by calculating the mean of 13 of 15 of the QLQ-C30 scales (global QoL and financial impact scales are excluded) after reversing the symptom scales [61]. This resulted in a metric variable. However, as not all variables were normally distributed, we used non-parametric tests to determine correlations. IBM SPSS Statistics version 28.0.0.0 was then used for statistical analysis, determining correlations between scores using Spearman’s rho correlation coefficient. The means of the SMI were compared through an independent sample *t* test. 

## 3. Results

A total of *n* = 80 patients took part in this study. Of those patients, 35 had not yet received treatment and were thus assigned to the t1 chemo-naïve group. A total of 45 patients had already received chemo- or radiation therapy and were assigned to the t2 post-neoadjuvant treatment group. 

For general study cohort information, please see Table 1, Table 2 and Table 3. 

### 3.1. Nutritional Scores

The nutritional scores used in this study all had strong intercorrelation (all *p*-values *p* < 0.001); the highest correlation coefficients in both groups were shown for MNA and MUST for the t1 group (Spearman’s rho coefficient: 0.791) and MNA and NRS for the t2 group (Spearman’s rho coefficient: 0.796).

### 3.2. HRQoL Summary Score

The observed summary scores were slightly lower in the t2 group in comparison to the t1 group, but no great difference was shown in the observed time points (t1: median = 80.4; t2: median = 74.6) (see Figure 2). 

The summary scores before chemotherapy (HRQoL 1) correlated significantly with all nutritional scores prior to chemotherapy, mainly the MNA (see Figure 2) and NRS scores (HRQoL1 × MNA1 *p* < 0.001; HRQoL1 × MUST *p* = 0.021; HRQoL1 × NRS1 *p* < 0.001), as well as physical functioning (*p* < 0.001), global QoL (*p* = 0.007) and SMI before chemotherapy (HRQoL1 × SMI1 *p* < 0.001, see Figure 3a). The summary scores after chemotherapy did not correlate with their respective SMI values (see Figure 3b); however, they did correlate with physical function (*p* < 0.001) and the global QoL score (*p* = 0.003).

### 3.3. Skeletal Muscle Index

Similarly to the HRQoL, skeletal muscle was shown to be similar before and after neoadjuvant treatment, with a mean of 48.38 cm^2^/m^2^ after chemotherapy from 49.65 cm^2^/m^2^ prior to chemotherapy, and the SMI before chemotherapy strongly correlated with the SMI after chemotherapy (SMI1 × SMI2 *p* < 0.001; see Figure 4). The SMI also correlated with physical function and the global QoL score prior to chemotherapy (PF2: *p* < 0.001; see Figure 5a; QL2 *p* = 0.049); however, after chemotherapy, this correlation could not be established (*p* = 0.734; see Figure 5b). 

Furthermore, in the t1 group, the HRQoL summary scores obtained before chemotherapy correlated with the SMI after chemotherapy (HRQoL1 × SMI2 *p* < 0.001).

## 4. Discussion

Sarcopenia and HRQoL are established risk factors for poor survival and tumor growth in upper GI tumors [10,12,19,41,62,63,64,65,66,67]. Sarcopenia has traditionally been measured through the CT-graphically obtained SMI, while HRQoL is obtained through patient questionnaires. While the assessment of the SMI is a rather elaborate process, the assessment of nutritional risk scores or HRQoL only requires minimal involvement of the physician and can be performed in a point-of-care manner. Physicians might obtain a valid and quick assessment of patients’ physical fitness and risk for sarcopenia through the assessment of patients’ HRQoL. 

As such, figuring out relations between HRQoL and sarcopenia might provide the key to simplifying the acquisition of outcome predictors. However, associations between sarcopenia and HrQoL have not been assessed widely. A study by Zou et al. showed that in gastric cancer, sarcopenia at baseline is an independent risk factor for postoperative complications and poor HRQoL 6 months after surgery [67]. Similarly, a study by Nipp et al. found that patients with cancer with sarcopenia reported lower HRQoL than their counterparts [24]. With these ideas in mind, our aim was to assess connections between physical fitness as indicated by patients, HRQoL, nutritional risk scores and measured skeletal muscle mass.

In our study, HRQoL was shown to be significantly associated with nutritional scores and the CT-graphically measured SMI. Prior to chemotherapy, there appears to be a correlation between SMI, nutritional scores and HRQoL, suggesting that these factors are associated at the initial diagnosis of upper GI tumors and, thus, HRQoL might allow for a ballpark estimation of nutritional status and muscle mass. This is supported by the results of recent studies that found an association of HRQoL and sarcopenia in patients with heart failure and patients with colorectal cancer [45,68]. Another recent study by Celoto et al. could not find a correlation between HRQoL and muscle mass in patients undergoing hemodialysis, which seems plausible as this condition differs vastly from cancer [69]. A study in patients with liver cell cancer, however, also found a correlation between sarcopenia and HRQoL in a prospective study [70]. Similarly, a recent study on machine learning used QoL as one of three factors to establish a prediction model of sarcopenia [71]. This association of sarcopenia and HRQoL is important, as the presence of sarcopenia is associated with adherence to prescribed chemotherapy regimens and the occurrence of dose-limiting toxicity during neoadjuvant chemotherapy [31,72,73]. Furthermore, impaired nutritional status as well as sarcopenia might lead to poor postoperative recovery, including high rates of anastomotic leakage, postoperative pneumonia and hospital readmittance, and impaired progress-free survival [31,32,33,34,35,39,40,47,54]. As a potential reason, a study by Dalamaga et al. suggested that low muscle mass can contribute to tumor growth through increased inflammation [74]. The correlation of nutrition and QoL found in our study is also supported by studies for patients with lung cancer, colorectal cancer and pancreatic cancer [13,14,75,76]. A study by Gharagozlian et al. on long-term survivors of gastrectomies for cancer found patients with malnutrition to have lower QoL, and a study by Maia et al. produced similar results [77,78].

However, HRQoL did not correlate with nutritional scores or SMI after chemotherapy. This is not surprising, as problems with eating such as dysphagia and the resulting malnutrition are often the primary symptom of upper GI tumors and, as such, are often the most prominent prior to diagnosis (and thus, the most impactful on HRQoL), while HRQoL after neoadjuvant treatment is often dominated by adverse effects of chemotherapy, with nutritional problems taking a backseat [26]. 

The disparity of HRQoL and SMI after chemotherapy could support a recent emphasis on the functional rather than quantitative assessment of skeletal muscle. A recent study by Barbosa et al. found good correlation with HRQoL for handgrip strength rather than SMI [45]. Contrarily to expectations and another study [43], the SMI remained relatively stable prior to and after chemotherapy. Alternatively, supportive nutritional and psychological therapy during neoadjuvant treatment might improve HRQoL to such a degree that they start overshadowing the loss of skeletal muscle in importance for HRQoL. Nonetheless, our findings support the emphasis on a comprehensive assessment of skeletal muscle including physical function or muscle quality over a purely quantitative assessment through SMI.

Recent studies have focused more on the assessment of muscle strength and function than on mass calculation [54,77,79,80]. For example, a 2020 study by Huang et al. and a 2022 study by Cai et al. used a combination of handgrip strength (HGS) and CT-assessed SMI to assess sarcopenia and showed it to be predictive of postoperative complications and long-term survival [32,54]. Similarly, a study by Chen et al. used a combination of CT scans, HGS and 6 m gait speed and showed similar results [81]. A study by Kurita found a better correlation of HGS with postoperative pneumonia than with SMI [80]. Another common method of assessment is a combination of HGS and bioelectrical impedance analysis [77]. Physical function furthermore appears to be associated with HRQoL in survivors of colorectal cancer, while this has not been assessed for tumors of the upper gastrointestinal tract [30]. Decreased physical activity appears to be linked to symptom-related problems like pain and appetite loss and socio-demographic factors [82].

Despite the loss of correlation after chemotherapy as described above, the SMI and HRQoL obtained prior to chemotherapy correlated with their corresponding post-chemotherapy scores. HRQoL prior to chemotherapy also correlated with the SMI after chemotherapy. Thus, physicians might obtain a valid and quick assessment of patients’ physical fitness and post-chemotherapy risk for sarcopenia through the assessment of patients’ HRQoL prior to chemotherapy. Further research is needed to isolate HRQoL as a predictor of post-chemotherapy and maybe also postoperative sarcopenia. 

As a limitation of this study, we did not investigate the effect of physical fitness, nutritional scores or HRQoL on cancer-specific or long-term survival. Further studies would be needed to assess this important connection. Additionally, due to the relatively low incidence of the disease in Germany, our number of included patients is rather small. A further limitation is that the patients were mostly older patients with additional comorbidities such as diabetes and arterial hypertension. Those prior conditions may additionally influence HRQoL and, as such, render the results of this study not applicable to healthy populations. Additionally, this study was conducted under COVID-19 restrictions. Due to this, some QLQ-C30 scores such as social functioning and role functioning may be not accurate; however, most of the scores that comprise the summary score would not be influenced by restrictions such as social distancing. Taking this into consideration, we think the results of this study were not greatly impacted by the pandemic.

However, given the abovementioned reservations on the comprehensive usage of the quantitative measurement of muscle mass, we provide data on the important relationship between patients’ HRQoL, nutritional status and measured muscle mass as an indicator for sarcopenia. The clinical implication of our findings is reasonable: A quick assessment of HRQoL and nutritional status at first presentation in the outpatient clinic without the requirement of any technical devices might lead to an individual, patient-centered journey through multimodal cancer therapy. 

## 5. Conclusions

Muscle mass, nutritional status and HRQoL (especially physical function) are intertwined parameters that can be used for the assessment of sarcopenia in the therapy of patients with tumors of the upper gastrointestinal tract. The different nutritional scores correlated well at all time points, while correlation with HRQoL and sarcopenia was best prior to chemotherapy. Initial patient-reported HRQoL also correlated well with the level of sarcopenia after chemotherapy and can therefore be used to assess challenges in the therapeutic course. An early, quick assessment of HRQoL and nutritional status is easily performed and paves the way for individual, holistic cancer therapy.

Further studies should be conducted on the relationship between muscle mass and HRQoL using newer methods of sarcopenia assessment such as bioelectrical impedance analysis, ideally in a multi-center setting. Furthermore, the results of this study warrant further studies to investigate whether the observed results would change with controlled dietary interventions. 

## Figures and Tables

**Figure 1 cancers-16-01232-f001:**
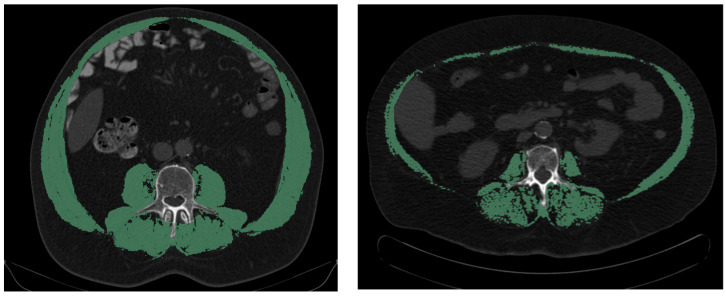
Semi-automatic measurement of skeletal muscle index (SMI) with the software 3D Slicer. On the left side is an example of a patient with a high SMI (75.5 cm^2^/m^2^), and on the right side is an example of a patient with a low SMI (38.3 cm^2^/m^2^).

**Figure 2 cancers-16-01232-f002:**
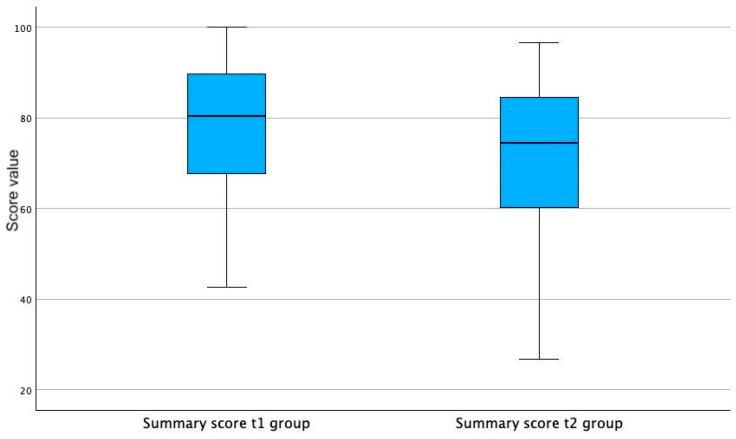
Boxplots of QLQ-C30 summary score at t1 and at t2.

**Figure 3 cancers-16-01232-f003:**
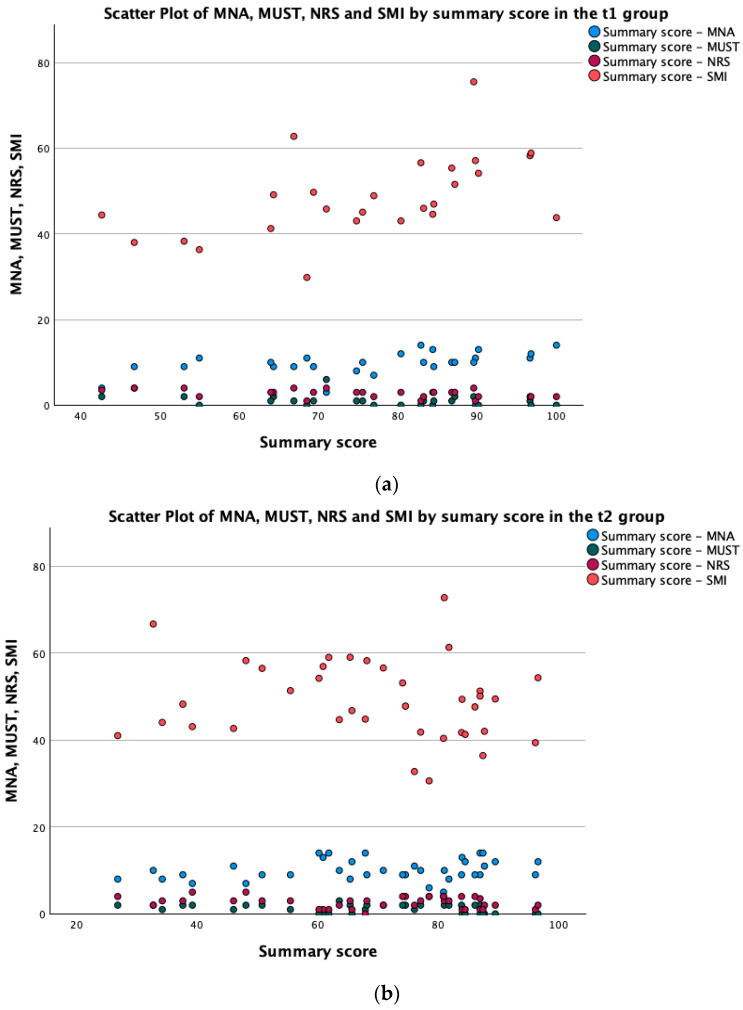
(**a**,**b**) Scatter plot of MNA, SMI, NRS and MUST by summary score at (**a**) t1 and (**b**) t2.

**Figure 4 cancers-16-01232-f004:**
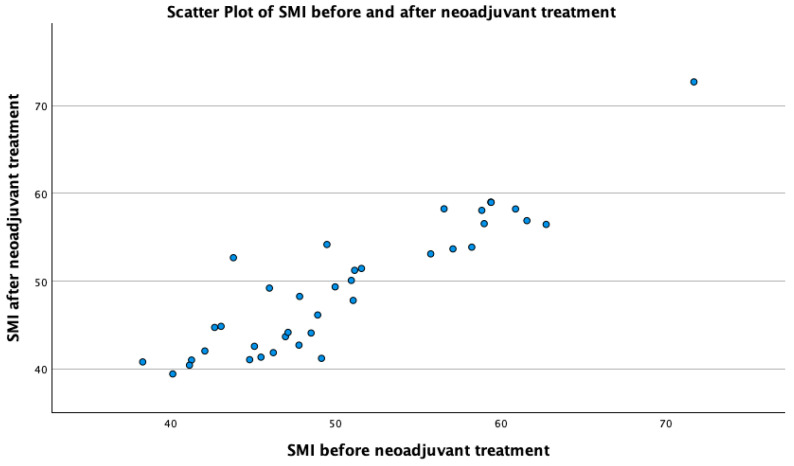
Scatter plot of skeletal muscle index (SMI) before and after neoadjuvant treatment.

**Figure 5 cancers-16-01232-f005:**
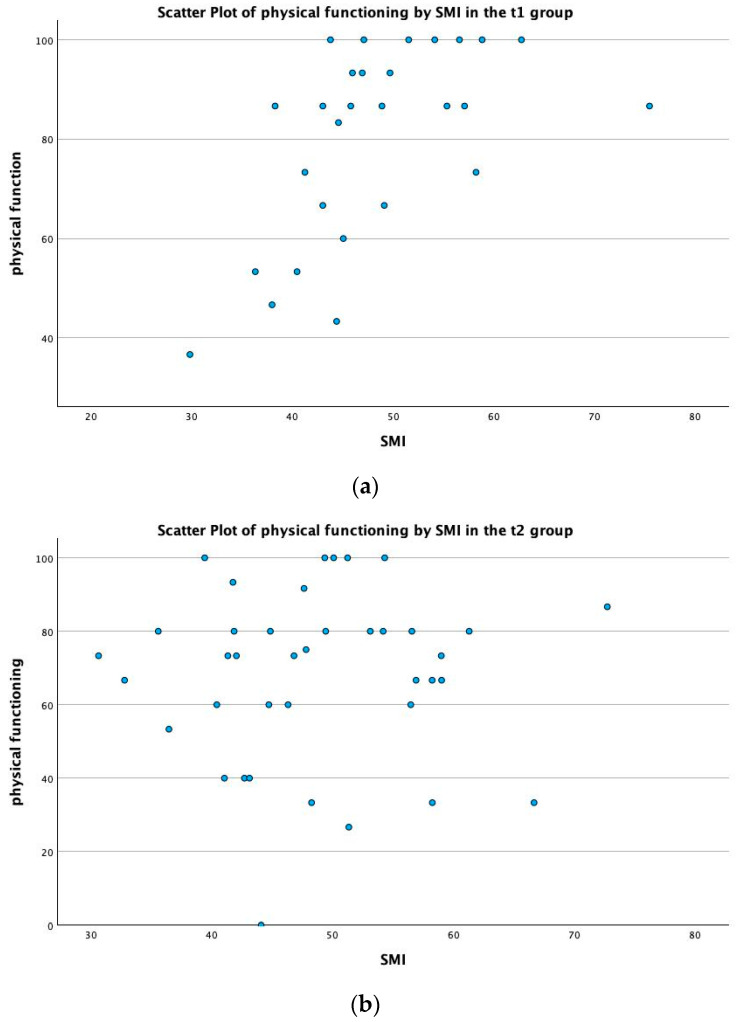
(**a**,**b**) Scatter plot of physical functioning and skeletal muscle index (SMI) before and after neoadjuvant therapy (at t1 and t2).

**Table 1 cancers-16-01232-t001:** Patient characteristics. Abbreviations: EG junction: esophagogastric junction; ASA: American Society of Anesthesiologists; SMI: skeletal muscle index; MNA: Mini Nutritional Assessment; MUST: Malnutrition Universal Screening Tool; NRS. Nutritional Risk Screening; QL2: quality of life; PF2: physical functioning; Sum: summary score; SD: Standard Deviation.

		N (%)	Median (IQR)	Min	Max
Age			65.0 (19.0)	28.0	82.0
Height in cm			175.0 (10.0)	147.0	194.0
Weight in kg			80.0 (18.4)	38.0	141.0
Tumor localization	EG junction	41 (51.2)			
	esophageal	19 (23.8)			
	gastric	20 (25.0)			
Sex	female	14 (17.5)			
	male	66 (82.5)			
ASA	ASA ≤ 2	22 (27.5)			
	ASA ≥ 3	58 (72.5)			
Weight loss in %			6.6 (13.8)	0.0	47.1

**Table 2 cancers-16-01232-t002:** Patient characteristics prior to neoadjuvant therapy. Abbreviations: SD: Standard Deviation; SMI: skeletal muscle index; MNA: Mini Nutritional Assessment; MUST: Malnutrition Universal Screening Tool; NRS: Nutritional Risk Screening; QL2: QoL; PF2: physical functioning; Sum: summary score.

		N (%)	Median (IQR)	Min	Max
Physical performance (measured in number of floors patients were able to climb)	≤1	6 (17.1)			
	2–3	9 (25.7)			
	>3	20 (57.1)			
SMI in cm^2^/m^2^			48.14 (13)	29.8	75.5
MNA	0–7	4 (12.1)			
	8–11	18 (54.6)			
	12–14	11 (33.3)			
	Missing	2			
MUST	0	15 (44.1)			
	1	10 (29.4)			
	≥2	9 (26.5)			
	Missing	1			
NRS	<3	15 (45.5)			
	≥3	18 (54.5)			
	Missing	2			
QL2			58.3 (25.0)	25.0	100.0
PF2			86.7 (33.3)	33.3	100.0
Sum			80.4 (23.0)	42.6	100.0

**Table 3 cancers-16-01232-t003:** Patient characteristics after neoadjuvant therapy. Abbreviations: SMI: skeletal muscle index; MNA: Mini Nutritional Assessment; MUST: Malnutrition Universal Screening Tool; NRS: Nutritional Risk Screening; QL2: quality of life; PF2: physical functioning; Sum: summary score.

		N (%)	Median (IQR)	Min	Max
Physical performance (measured in number of floors patients were able to climb)	≤1	11 (26.2)			
	2–3	9 (21.4)			
	>3	22 (52.4)			
	Missing	3			
SMI in cm^2^/m^2^			47.7 (12.0)	30.6	72.7
MNA	0–7	6 (14.3)			
	8–11	24 (57.1)			
	12–14	12 (28.6)			
	Missing	3			
MUST	0	13 (31.0)			
	1	5 (11.9)			
	≥2	24 (57.1)			
	Missing	3			
NRS	<3	18 (43.9)			
	≥3	23 (56.1)			
	Missing	4			
QL2			50.0 (33.3)	16.7	100.0
PF2			73.3 (29.2)	0.0	100.0
Sum			74.6 (27.0)	26.8	97.0

## Data Availability

Data are available upon reasonable request from the corresponding author.

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
