# Peer review of "Quality of Life, Sarcopenia and Nutritional Status in Patients with Esophagogastric Tumors before and after Neoadjuvant Therapy"

_cancers, 2024, doi:10.3390/cancers16061232_

Round 1

Reviewer 1 Report

Comments and Suggestions for Authors

This is an interesting study with adequate novelty and quality. However, some points should be addressed.

-The 1st and the 3 paragrpahs of the Introduction section is too short. the authors should add further data concenring their topic by simultaneously adding relevant references.

- The introduction shoulf be enhriched by adding a separate paragraph about the impact of Quality of Life, Sarcopenia and Nutritional Status in Patients with other types of cancer.

-  At the Methods section, the authors should specify if their data fron the questionnaires are self-reported or they performed a face-to face interview with the patients.

- The statistical analysis section is quite compex. The authors should report firstly the type of variables used (e.g., categorical, cinunious, etc) and the what statistical test they used according to the nature of their variables. What test they used to evaluate whether their variables follow  on not the normal distribution. In case of non-normally distributed variables, non parametric statistical test must be used, reporting a median and not an mean value for each variable.

- The abbreviations of the Tables should be removed at the end of them.

- The name of the variable of the y axis is missing in Figure 2.

- Figure 4 needs improvement concerning its resolution.

- The 1st paragraph of the Discussion sections needs the addition of some updated references from the last 2-3 years.

-  The 2nd and the 3rd paragraphs of the discussion section should be in one paragraph.

- In lines 215-216 of the Discussion section, the authors should report in more details the results of studies with reference numbers 13, 20, 21, 22.

- In general, the discussion section need further improvement by comparing previous relevant studies with the results found in the present study.

- In the conclusion section, the uthors should add some statements concerning what future studies could be performed based on the results of their study.

- In section 2.1, the authors did not report as an exclusion criteria any comorbidity. Thus, if some of th patients under study had any comorbidity, this should be reported in the limitation of the study.

- The mean age of the patients is almost 65 years, ehich means that most of the patients were older adults. This should be reported in the limitations of the study.

- The study was performed during the period of Covid-19 pandemic. The authors should report this in the discussion section reporting whether this specific condition may affect their results.

Comments on the Quality of English Language

Moderate editing of English language required

Author Response

This is an interesting study with adequate novelty and quality. However, some points should be addressed.

Thank you for taking the time to read our manuscript and adding valuable comments.

-The 1st and the 3 paragrpahs of the Introduction section is too short. the authors should add further data concenring their topic by simultaneously adding relevant references.

- The introduction shoulf be enhriched by adding a separate paragraph about the impact of Quality of Life, Sarcopenia and Nutritional Status in Patients with other types of cancer.

Thank you for your comments. We expanded the first paragraph with comprehensive information on sarcopenia and HRQoL, and their relevance in other types of cancer, effectively providing a more complete impression of the importance of studying their impact in gastric cancer. Meanwhile, we moved the 3rd paragraph to the method section to clarify why we chose the selected scores for analysis in our study. We hope that the introduction is now more complete and to your satisfaction.

-  At the Methods section, the authors should specify if their data fron the questionnaires are self-reported or they performed a face-to face interview with the patients.

Thank you for your comment. The questionnaires are self-reported and we clarified this in the methods section.

- The statistical analysis section is quite compex. The authors should report firstly the type of variables used (e.g., categorical, cinunious, etc) and the what statistical test they used according to the nature of their variables. What test they used to evaluate whether their variables follow  on not the normal distribution. In case of non-normally distributed variables, non parametric statistical test must be used, reporting a median and not an mean value for each variable.

Thank you for your comment. This was an oversight on our part. As not all of the variables are normally distributed, we changed  to a non-parametrical test to test accordingly. The results however and thus the overall message remained the same. Concerning the question of means or medians for the two sections where we compared QoL and SMI for groups 1 and 2, the SMI was normally distributed, so this section remained as it was before while we changed the QoL accordingly to clarify that this is a purely descriptive observation. In the tables, we changed mean to median for all variables to avoid confusion and aid in the comprehension of the tables.

- The abbreviations of the Tables should be removed at the end of them.

- The name of the variable of the y axis is missing in Figure 2.

- Figure 4 needs improvement concerning its resolution.

Thank you very much for your comments, we amended our manuscript accordingly and hope this is now to your satisfaction.

- The 1st paragraph of the Discussion sections needs the addition of some updated references from the last 2-3 years.

-  The 2nd and the 3rd paragraphs of the discussion section should be in one paragraph.

- In lines 215-216 of the Discussion section, the authors should report in more details the results of studies with reference numbers 13, 20, 21, 22.

- In general, the discussion section need further improvement by comparing previous relevant studies with the results found in the present study.

Thank you for your comments. We expanded the discussion with further references and discussion of relevant studies and hope this is now to your satisfaction. Additionally there seems to have been a formatting error during our submission process so the references were out of order. We corrected this and apologize for the possible confusion.

- In the conclusion section, the uthors should add some statements concerning what future studies could be performed based on the results of their study.

- In section 2.1, the authors did not report as an exclusion criteria any comorbidity. Thus, if some of th patients under study had any comorbidity, this should be reported in the limitation of the study.

- The mean age of the patients is almost 65 years, ehich means that most of the patients were older adults. This should be reported in the limitations of the study.

- The study was performed during the period of Covid-19 pandemic. The authors should report this in the discussion section reporting whether this specific condition may affect their results.

Thank you for your comments. We expanded these sections to include your suggestions and hope they are satisfactory.

Reviewer 2 Report

Comments and Suggestions for Authors

With this study, the researchers set themselves a very specific aim, to highlight the quality of life in those affected by upper GI neoplasia. The number of patients enrolled is not high, but the study years are few and this makes the study group more homogeneous. The results are absolutely acceptable, but we ask some questions:

1 Was the questionnaire they administered to the patients self-administered or read and filled in by a doctor/nurse?

In the past we have conducted similar work with self-administration (PMID: 9617108 to be cited in the bibliography) in order to avoid external interference which would have partially nullified the survey.

2 linking sarcopenia with quality of life seems correct to us and certainly gives more reliable results, but why measure muscles with CT scan and not with skinfold measurement? Has the weight of the patients also been taken into account? Have you made use of the collaboration of a dietician?

3 certainly for all neoplastic pathologies we will arrive at personalized therapies, and we agree on this, but the final sentence does not seem appropriate to me for this paper.

The iconography is appropriate, the English is good, the bibliography supports the conclusions.

Author Response

With this study, the researchers set themselves a very specific aim, to highlight the quality of life in those affected by upper GI neoplasia. The number of patients enrolled is not high, but the study years are few and this makes the study group more homogeneous. The results are absolutely acceptable, but we ask some questions:

First of all, thank you for taking the time to read and revise our manuscript. You added valuable points and we hope we were able to address them to your satisfaction.

1 Was the questionnaire they administered to the patients self-administered or read and filled in by a doctor/nurse?

In the past we have conducted similar work with self-administration (PMID: 9617108 to be cited in the bibliography) in order to avoid external interference which would have partially nullified the survey.

Thank you very much for your question and your suggestion for further references. The questionnaire was self-administered. In the past we also conducted a project on the concordance of physician-administered versus self-administered questionnaires (that has not yet been published) which showed the concordance to be relatively low. As you said, in such a case the external interference would severely impact the validity of a study, we cited you accordingly.

2 linking sarcopenia with quality of life seems correct to us and certainly gives more reliable results, but why measure muscles with CT scan and not with skinfold measurement? Has the weight of the patients also been taken into account? Have you made use of the collaboration of a dietician?

Thank you for your question. Assessment of muscle mass with CT scan was a method previously established in our department and thus seemed natural for this study aim. Furthermore, we believe CT scans to be the most reliable and reproduceable method for determining SMI. For SMI calculated by CT images, the muscle mass is divided by squared height and thus not adapted for weight. However, normal values for SMI exist both for methods that adapt for weight and height. It would certainly be interesting to also use another method for assessing muscle mass that adjusts for weight, suchs as bioelectrical impedance analysis, and set more focus on body composition and muscle quality. We have since established these methods in our department and hope to use them in future studies.
Concerning the collaboration of a dietician- patients that express difficultied with eating or significant weight loss are routinely sent for consultations with our in-house dieticians as a part of their treatment. For this study, we adhered to standard prehabilitation protocol in our department.

3 certainly for all neoplastic pathologies we will arrive at personalized therapies, and we agree on this, but the final sentence does not seem appropriate to me for this paper.

Thank you for your comments, we amended this part and hope it is now to your satisfaction. We want to stress that we did not mean for this sentence to refer to personalized or immunotherapy, but rather to a more personalized therapy on regards to placing more importance on the individual patient’s needs and constitution. We are sorry that our wording left place for misunderstandings

The iconography is appropriate, the English is good, the bibliography supports the conclusions.

Round 2

Reviewer 1 Report

Comments and Suggestions for Authors

The authors have significantly improved their manuscript by addressing all my suggestions.

Comments on the Quality of English Language

 Minor editing of English language required